# New Taxa of the Family Amniculicolaceae (Pleosporales, Dothideomycetes, Ascomycota) from Freshwater Habitats in Spain

**DOI:** 10.3390/microorganisms8091355

**Published:** 2020-09-04

**Authors:** Viridiana Magaña-Dueñas, Alberto M. Stchigel, José F. Cano-Lira

**Affiliations:** Mycology Unit, Medical School and IISPV, Universitat Rovira i Virgili, Sant Llorenç 21, 43201 Reus, Tarragona, Spain; qfbviry@hotmail.com (V.M.-D.); jose.cano@urv.cat (J.F.C.-L.)

**Keywords:** Ascomycota, freshwater, fungi, plant debris, *Pleosporales*, Spain

## Abstract

With the exception of the so-called Ingoldian fungi, the diversity and distribution of the freshwater aero-aquatic or facultative fungi are not well known in Spain. In view of that, we collected and placed into wet chambers 105 samples of submerged and decomposing plant debris from various places in Spain, looking for individuals belonging to these latter two morpho-ecological groups of fungi. As a result, we found and isolated in pure culture several fungi, the morphology of some of them belonging to the family *Amniculicolaceae* (order *Pleosporales*, class *Dothideomycetes*). After a careful phenotypic characterization and a phylogenetic tree reconstruction using a concatenated sequence dataset of D1-D2 domains of the 28S nrRNA gene (*LSU*), the internal transcribed spacer region (ITS) of the nrDNA, and a fragment of the translation elongation factor 1-alpha (*tef1*) gene, we report the finding of three new species of the genus *Murispora*: *Murispora navicularispora*, which produces cinnamon-colored, broadly fusiform to navicular ascospores; *Murispora fissilispora*, which has as a remarkable characteristic the production of both sexual and asexual morphs in vitro; and *Murispora asexualis*, the unique species of the genus that lacks a sexual morph. As a consequence of the phylogenetic study, we introduce the new aero-aquatic genus *Fouskomenomyces*, with a new combination (*Fouskomenomyces cupreorufescens*, formerly *Spirosphaera cupreorufescens* as the type species of the genus) and a new species, *Fouskomenomyces mimiticus*; we propose the new combinations *Murispora bromicola* (formerly *Pseudomassariosphaeria bromicola*) and *Murispora triseptata* (formerly *Pseudomassariosphaeria triseptata*); and we resurrect *Massariosphaeria grandispora*, which is transferred to the family Lopiostomataceae.

## 1. Introduction

Fungi are a diverse group of ubiquitous organisms present in almost all ecosystems on Earth, including aquatic habitats [1]. Several fungal taxa have been isolated from freshwater environments, which offer a wide range of organic substrates for fungal colonization [2]. The most important role of fungi in freshwater is the recycling of dead organic matter, typically submerged plant debris [3]. Freshwater fungi complete (at least one part of) their life cycle into the water, and disperse their propagules through the water. Freshwater fungi are generally classified into different sorts of morphological and ecological groups: the “Ingoldian”, producing submerged star-like (*stauro*-) or worm-like (*scoleco*-) asexual spores (or propagules) in lotic habitats (moving waters) [2,4]; the aero-aquatic, forming helical, net-like or globose conidia above the surface of lentic (standing) waters [5]; and members of the Ascomycota reproducing by the formation of conidia or sexual propagules (ascospores) into fertile bodies (conidiomata and ascomata, respectively). Members of the Ascomycota reproducing by ascospores seem to be less exclusively adapted to life in aquatic environments than the other sort of previously cited fungal groups [6]. These fungi produce unitunicate asci with apical structures, or these are fissitunicate (bitunicate) and the ascospores are mostly ornate with mucilaginous sheaths or appendages which facilitate the attachment to submerged substrates [7].

The freshwater Ascomycota (FWA) are one of the least studied groups of fungi but they are taxonomically diverse and have representatives in a wide spectrum of families and orders. Approximately one third of the FWA with sexual reproduction belongs to the class *Dothideomycetes* [8]. Four lineages of the class *Dothideomycetes* have been recently described from this habitat: the order *Jahnulales* [9], the family *Lindgomycetaceae* [10,11], the family *Natipusillaceae* [12] and the family *Amniculicolaceae* [13,14]. The latter was established by Zhang in 2009 [15] to accommodate the genera *Amniculicola, Murispora* and *Pseudomassariosphaeria* [15]. Although all species of those three genera grow and produce a purple pigment on submerged wood, they differ in the morphology of the ascospores, which vary in color (hyaline in *Amniculicola*, and brown in *Murispora* and in *Pseudomassariosphaeria*) and septation (1-septate in *Amniculicola,* transversely multiseptate in *Pseudomassariosphaeria*, and with multiple transversal, longitudinal and oblique septa (muriform) in *Murispora*) [15,16]. Most of the *Amniculicolaceae* have been reported from freshwater habitats in Italy, France, Germany, Denmark and China [15,16,17,18].

During a survey on fungi living on decaying plant material in freshwater habitats in Spain, several strains that are morphologically compatible with members of the family *Amniculicolaceae* have been isolated in pure culture. The objectives of this study were to characterize phenotypically and to identify such fungi by phylogenetic analysis using nucleotide sequences of informative molecular markers.

## 2. Materials and Methods

### 2.1. Sample Collection and Specimen Examination

A total of 105 samples of submerged plant material were collected: three in *Les Guilleries* (Barcelona province), 50 in *Cascadas del Huéznar* (Cazalla de la Sierra, Sevilla province), 22 in Capafonts (Tarragona province) and 30 in *Serra del Montsant* (Tarragona province), Spain. These were placed into sterile self-sealing plastic bags to be transported to the lab and stored until processing. The specimens were rinsed twice with tap water, placed into Petri dishes or appropriate plastic containers lined inside with filter paper and moistened with sterile water with diehldrin^®^ (20 drops of a solution of 20 mg diehldrin in 20 mL of dimethyl-ketone/L of water), incubated at room temperature (20–25 °C), and examined periodically under stereomicroscope for up to 2 months. Several ascomata or asexual propagules were taken and transferred using sterile disposable tuberculin-type needles to 55 mm diam. Petri dishes containing oatmeal agar (OA; 30 g of filtered oat flakes, 15 g agar-agar, 1 L tap water [19]), then incubated at room temperature. All isolates were stored in the culture collection of the Faculty of Medicine at Universitat Rovira i Virgili (FMR; Reus, Spain). Type specimens and ex-type cultures of the novel fungi were deposited in the Westerdijk Fungal Biodiversity Institute (CBS), Utrecht, The Netherlands (Table 1).

### 2.2. Phenotypic Study

Macroscopic characterization of the colonies was performed on OA, 2% malt extract agar (MEA; Difco, Detroit, MI, USA) [19] and potato dextrose agar (PDA; Pronadisa, Madrid, Spain) [20] into 90 mm diam. Petri dishes, after incubation for three weeks at 15 °C in the dark for species of the genus *Murispora* [16], and in similar conditions but at 20 °C for other taxa. Color notations were according to Kornerup and Wanscher (1978) [21]. The ability of the isolates to grow at cardinal temperatures was determined on PDA after 7 d in the dark, ranging from 5 to 35 °C, at 5 °C intervals, but also at 37 °C. Measurements and descriptions of microscopic structures were taken from specimens mounted in Shear’s mounting medium (3 g potassium acetate, 60 mL glycerol, 90 mL ethanol 95% and 150 mL distilled water) [22], using an Olympus BH-2 bright field microscope (Olympus Corporation, Tokyo, Japan). Photomicrographs were taken using a Zeiss Axio-Imager M1 microscope (Oberkochen, Germany) with a DeltaPix Infinity × digital camera using Nomarski differential interference contrast.

### 2.3. DNA Extraction PCR Amplification and Sequencing

The strains were cultured on PDA for 7 days at 25 °C in the dark. Total DNA was extracted using the FastDNA kit protocol (Bio101, Vista, CA, USA), with a FastPrep FP120 instrument (Thermo Savant, Holbrook, NY, USA) according to the manufacturer’s protocol. DNA was quantified by using Nanodrop 2000 (Thermo Scientific, Madrid, Spain). The following *loci* were amplified and sequenced: *LSU* (28S nrRNA gene), with the primer pair LR0R [23] and LR5 [24]; ITS (internal transcribed spacer region), with the primer pair ITS5 and ITS4 [25]; and *tef1* with EF1-983F and EF1-2218R [26]. The PCR amplifications were performed in a total volume of 25 µL containing 5 µL 10 × PCR Buffer (Invitrogen, CA, USA), 0.2 mM dNTPs, 0.5 µL of each primer, 1 U Taq DNA polymerase and 1–10 ng genomic DNA. PCR conditions for *LSU* and ITS were set as follows: an initial denaturation at 95 °C for 5 min, followed by 35 cycles of denaturation, annealing and extension, and a final extension step at 72 °C for 10 min. For the *LSU* and ITS amplification, the 35 cycles consisted of 45 s at 95 °C, 45 s at 53 °C and 2 min at 72 °C; and for the *tef1* an initial denaturation at 94 °C for 2 min, followed by 30 cycles consisting of 30 s at 94 °C, 1 min 20 s at 57 °C and 1 min 30 s at 72 °C. PCR products were purified and stored at −20 °C until sequencing. The same pairs of primers were used to obtain the sequences at Macrogen Spain (Macrogen Inc., Madrid, Spain). The consensus sequences were obtained using the SeqMan software v. 7 (DNAStar Lasergene, Madison, WI, USA).

### 2.4. Phylogenetic Analysis

The sequences generated in this study were compared with those of the National Center for Biotechnology Information (NCBI) using the Basic Local Alignment Search Tool (BLAST; https://blast.ncbi.nlm.nih.gov/Blast.cgi). Alignment for each *locus* was performed with the MEGA (Molecular Evolutionary Genetics Analysis) software v. 7.0. [27], using the ClustalW algorithm [28] and refined with MUSCLE [29] or manually, if necessary, on the same platform. The alignment included our sequences, together with those available at the NCBI databases, of all genera and species belonging to the family *Amniculiculaceae*, and representatives of the families *Lindgomytaceae, Teratospharriaceae, Lophiostomataceae* and *Sporormiaceae* (Table 1). The phylogenetic analyses were carried out using Maximum-Likelihood (ML) and Bayesian Inference (BI) with RAxML v. 8.2.10 [30] using the Cipres Science gateway portal [31] and MrBayes v. 3.2.6 [32], respectively. For ML analyses, the best nucleotide substitution model was General Time Reversible with Gamma distribution. Support for internal branches was assessed by 1000 ML bootstrapped pseudoreplicates. For the BI phylogenetic analysis, the best nucleotide substitution model was determined using jModelTest [33]. For ITS, we used the symmetrical model with gamma distribution (SYM + G), for *LSU*, we used the symmetrical model with proportion of invariable sites and gamma distribution (SYM + I + G), and for *tef1*, we used the General Time Reverse with proportion of invariable sites and gamma distribution (GTR + I + G). The parameter settings were two simultaneous runs of 5M generations, four Markov chain Monte Carlo (MCMC), sampled every 1000 generations. The 50% majority-rule consensus tree and posterior probability values were calculated after discarding the first 25% of the samples. *Leptosphaeria dolium* (CBS 125,979 and CBS 505.75) served as outgroup taxa. Confident branch support was defined as Bayesian posterior probabilities (PP) ≥ 0.95 and ML bootstrap support (BS) ≥ 70%. Sequences generated in this study were deposited in European Nucleotide Archive (ENA).

## 3. Results

### 3.1. Phylogenetic Analyses

The final concatenated ITS-*LSU*-*tef1* sequence dataset using both ML and Bayesian analyses contained 37 ingroup strains from five families (*Amniculicolaceae*, *Lindgomytaceae*, *Lophiostomataceae*, *Sporormiaceae* and *Teratospharriaceae*). The alignment comprised a total of 1936 characters including gaps (815 for *LSU*, 399 for ITS and 722 for *tef1*), of which 435 were parsimony informative (125 for *LSU*, 173 for ITS and 137 for *tef1*). The individual sequence datasets did not show any conflicts in the tree topologies for the 70% reciprocal bootstrap trees, which allowed the three genes for the multi-locus analysis to be combined. The ML analysis showed similar tree topology and was congruent with that obtained in the Bayesian analysis. For the BI multi-locus analysis, a total of 2706 trees were sampled after the burn-in with a stop value of 0.01. The support values were slightly different with the two analysis methods: with BI, posterior probabilities being higher than the ML bootstrap support values. In our phylogenetic analysis, the family *Amniculicolaceae* formed a well-supported main clade (99% BS/1 PP) (Figure 1). All taxa in this family were split into two well-supported clades. The first one (99% BS/1 PP) included two accepted genera (*Amniculicola*, 96% BS/1 PP and *Vargamyces*, 91% BS/1 PP), plus another genus (81% BS/0.98 PP). We propose this one as the new *Fouskomenomyces*, comprising *Fouskomenomyces cupreorufescens* (basionym *Spirosphaera cupreorufescens*) and two of our strains (FMR 17,151 and FMR 16,958). The second main clade, corresponding to the genus *Murispora* (94% BS/1 PP), was represented by all previously described species (including the type species of the genus, *M. rubicunda*). Our strains FMR 17,248, FMR 17,251 and FMR 17,838 were placed into independent terminal branches, each one representing a new species for the genus and two new combinations, *M. bromicola* (basionym *Pseudomassariosphaeria bromicola*) and *M. triseptata* (basionym *P. triseptata*). Surprisingly, *Pseudomassariosphaeria grandispora* (formerly included in the *Amniculicolaceae*) fell into the *Lopiostomataceae* (96% BS/1 PP).

### 3.2. Taxonomy

*Amniculicolaceae* Y. Zhang ter, C.L. Schoch, J. Fourn., Crous & K.D. Hyde, Studies in Mycology 64: 95 (2009). MycoBank 515469.

*Type genus: Amniculicola* Y. Zhang ter & K.D. Hyde, Mycol. Res. 112(10):1189 (2008).

Because *Spirosphaera cupreorufescens* was placed into a terminal clade in the *Amniculicolaceae*, and the type species of the genus, *Spirosphaera floriformis*, is phylogenetically distant (in the class *Leotiomycetes* [34]), and our strains FMR 16,958 and FMR 17,151 grouped together on a sister branch in the same terminal clade as *S. cupreorufescens*, we erect the new genus *Fouskomenomyces*, and recognize two species: *Fouskomenomyces cupreorufescens* comb. nov. (the type species of the genus) and *Fouskomenomyces mimiticus* sp. nov.

***Fouskomenomyces*** V. Magaña-Dueñas, Cano & Stchigel, **gen. nov.** MycoBank MB835696.

*Etymology*. From Greek *φουσκωμένο*-, inflated, and -*μύκητα*, fungus, because of the nature of the propagules.

*Description: Mycelium* superficial to immersed composed by septate, smooth- and thin-walled, hyaline to pale brown, branching hyphae. *Conidiophores* micronematous to semi-macronematous, simple, pale brown, conidiogenous cells integrated, holoblastic, polyblastic. *Conidial propagules* brown to copper brown, more or less globose, scattered, composed by a compact branched system of globose to polyhedral cells, each one blown out successively to produce several daughter cells, detached by rhexolytic secession, or formed by branched, loosely spirally, interwoven septate filaments. *Chlamydospores* and *sexual morph* not observed.

*Type species: Fouskomenomyces cupreorufescens* (Voglmayr 2004) V. Magaña-Dueñas, Stchigel & Cano. MycoBank MB 835697.

***Fouskomenomyces cupreorufescens*** (Voglmayr 2004) V. Magaña-Dueñas, Stchigel & Cano, **comb. nov.** MycoBank MB 835697.

*Basionym*: *Spirosphaera cupreorufescens* Voglmayr, Studies in Mycology 50:221–228. (2004).

*Description*: Voglmayr (2004).

*Notes:* The main distinctive features of *F. cupreorufescens* are its production of coppery-brown conidia in mass, irregularly globose and up to 150 µm diam. and its branched, loosely spiralled, interwoven, septate filaments.

***Fouskomenomyces mimiticus*** V. Magaña-Dueñas, Cano & Stchigel, **sp. nov.** FMR 16,958. Mycobank MB 835698. (Figure 2)

*Etymology*. From Greek *μιμητικός*, mimetic, because the morphological resemblance to other genera such as *Pseudoagerita*.

*Description*: *Mycelium* superficial to immersed, composed by septate, smooth- and thin-walled, pale brown, branched, 2–3 μm wide hyphae. *Conidiophores* micronematous to semi-macronematous, simple, pale brown, conidiogenous cells integrated, polyblastic. *Conidial propagules* brown, globose to sub-globose, 55–150 μm diam., composed by a compact branched system of globose to polyhedral cells of 4–5 μm diam., each one successively budding out up 3–5 daughter cells, not breaking up into fragments when old, and detaching from the hyphae by rhexolytic secession. *Chlamydospores* and *sexual morph* absents.

Culture characteristics (after 3 weeks at 20 °C): Colonies on natural substratum not evident, appearing as scattered propagules. Colonies on MEA 2% reaching 25 mm diam., velvety, umbonate, margins regular, with abundant aerial mycelium, orange white to brownish-orange (6C4); reverse dark brown to brown (7F8/7E8), orange white (6C4) at the margins. Colonies on OA reaching 34–36 mm diam., flattened, slightly floccose, margins regular, with sparse aerial mycelium, dark purple to purplish grey (14F6/14B2); reverse violet to grey (15E8/14D1/15A3), margins white (1A1). Colonies on PDA reaching 28 mm diam., convex, cottony at the center, slightly floccose and velvety in the rest of the colony, margins regular, pinkish-white (9A2), margins orange grey (6B2); reverse brownish orange to reddish brown (6C3/8E7), margins orange white (5A2). Cardinal temperatures of growth: Optimum 20 °C, maximum 28 °C, minimum 15 °C.

*Material examined:* Spain, Barcelona province, Les Guilleries, from freshwater submerged plant debris, Nov. 2017, Eduardo Jose de Carvalho Reis, holotype CBS H-24461, culture ex-type FMR 16,958. Spain, Barcelona province, Les Guilleries, from freshwater submerged plant debris, Nov. 2017, Eduardo Jose de Carvalho Reis, living cultures FMR 17,151 = CBS 146935.

*Notes: Fouskomenomyces mimiticus* produces brown to dark brown, globose to sub-globose propagules, composed of a compact branched system of globose to polyhedral cells, whereas the propagules of *Fouskomenomyces cupreorufescens* are formed by branched, loosely spiralled, interwoven filaments, which are coppery-brown in mass.

Because the genus *Murispora* now includes three new species and two new combinations (Figure 1) displaying novel morphological features, we have amended it as follows:

*Murispora* Y. Zhang ter, J. Fourn. & K.D. Hyde, in Zhang et al., Stud. Mycol. 64: 95 (2009). MycoBank MB 515472.

Saprobic fungi living in freshwater habitats. *Ascomata* scattered or in small groups, immersed, erumpent, or nearly superficial, dark brown to black, ostiolate, globose to subglobose, neck periphysate with an apex weakly papillate, conical or nearly so. *Peridium* 3–7-layered, outer layer of *textura angularis* or *textura intricata*. *Pseudoparaphyses* trabeculate, embedded in mucilaginous material. *Asci* (4–)8-spored, bitunicate, fissitunicate, short pedicellate, cylindrical to clavate, with an ocular chamber. *Ascospores* transversally septate or muriform, hyaline when young, mostly becoming pale brown to reddish brown with age, less commonly remaining hyaline, constricted at the septa, navicular to broadly ellipsoidal, usually surrounded by an irregular, hyaline, gelatinous sheath. Staining the substrate in purple. *Asexual morph* coelomycetous.

*Type species: Murispora rubicunda* (Niessl) Y. Zhang ter, J. Fourn. & K.D. Hyde, in Zhang et al., Stud. Mycol. 64: 96 (2009).

≡ *Pleospora rubicunda* Niessl, Notiz. Pyr.: 31 (1876).

= *Massariosphaeria rubicunda* (Niessl) Crivelli, Ueber die heterogene Ascomycetengattung pleospora rabh.; Vorschlag für eine Aufteilung (Diss. Eid genössischen technischen hochschule Zürich 7318): 144 (1983).

= *Karstenula rubicunda* (Niessl) M.E. Barr, N. Amer. Fl., Ser. 2 (New York) 13: 52 (1990).

***Murispora bromicola*** (Phukhams., Ariyaw., Camporesi & K.D. Hyde) V. Magaña-Dueñas, Cano & Stchigel, **comb. nov.** MycoBank MB 835699.

*Basionym: Pseudomassariosphaeria bromicola* Phukhams., Ariyaw., Camporesi & K.D. Hyde, Ariyawansa et al., Fungal Diversity: 10.1007/s13225-015-0346-5, [2014] (2015).

Description: Ariyawansa et al. 2015.

*Notes:* Morphologically differing from the other species of *Murispora* by its production of hyaline ascospores (brown in the rest of the species of the genus), fusiform to lunate and narrower towards the apex (mostly ellipsoidal with rounded ends in other species), and not strongly constricted at the septa (although strongly constricted at septa in all other species of the genus).

*Pseudomassariosphaeria triseptata*, of marine origin, is a species recently introduced to the genus *Pseudomassariosphaeria* by Jones et al., in 2020 [35]. However, in our phylogenetic analysis, this species, as well as *P. bromicola*, was placed into the *Murispora* clade. Therefore, we propose the next new combination for this fungus.

***Murispora trisepata*** (E.B.G. Jones & Abdel-Wahab) V. Magaña-Dueñas, Cano & Stchigel, **comb. nov.** MycoBank MB 836493.

*Basionym: Pseudomassariosphaeria triseptata* E.B.G. Jones et Abdel-Wahab. Botanica Marina 63(2):157 (2020)

Description: Jones et al. 2020.

*Notes: Murispora triseptata* differs from all other species of the genus by possessing hyaline, 3-septate, ellipsoidal big ascospores [35].

Based on phenotypic features and phylogenetic results, three new species of *Murispora* are proposed as follows:

***Murispora fissilispora*** V. Magaña-Dueñas, Stchigel & Cano, **sp. nov.** FMR 17,251. MycoBank MB 835710 (Figure 3).

*Etymology*. From Latin *fissile*-, splitting, and -*sporarum*, spore, because the ascospores split at the middle when old.

*Mycelium* superficial to immersed, composed by septate, smooth- and thin-walled, pale brown, branched, 2–3 μm wide hyphae. *Ascomata* perithecial, immersed to semi-immersed, solitary, dark brown to black, ostiolate, papillate, *neck* conic-truncate, 105–108 × 60 μm, pyriform, 320–350 × 280–300 μm;, *peridial wall* 2–3-layered, 30–60 μm thick, outer wall of *textura intricata* composed of brown to dark brown hyphae 2–4 μm diam, inner wall layer hyaline and thin; *hamathecium* comprising numerous hyaline, filamentous, branched, septate *paraphyses* 1.5–2 μm wide;, periphysate; *asci* 4–8-spored, bitunicate, stipitate, cylindrical to cylindrical-clavate, 160–200 × 14–16 μm, stipe 20–25 μm long, without apical structures; *ascospores* hyaline when young, becoming brown at maturity, muriform, broadly fusiform to irregularly shaped, 15–27 × 6–8 μm, surrounded by a mucilaginous sheath, divided at the middle when old due to the narrowing of the medial septum. *Conidiomata* pycnidial, solitary, mainly immersed, pale brown to brown, ostiolate, subglobose, 65–70 × 85–90 μm; *conidiomata wall* of *textura angularis,* composed of pale brown to brown, flattened polygonal cells of 4–7 μm diam.; *conidiophores* reduced to the *conidiogenous cells*, which are phialidic, hyaline, smooth-walled, formed from the innermost layer of the pycnidial wall; *conidia* one-celled, hyaline, ovoid to ellipsoidal, 3–4 × 1.5–2.5 μm, guttulate.

Culture characteristics (3 weeks at 15 °C). Colonies on PDA reaching 20–22 mm diam., convex, floccose, margin regular, with abundant aerial mycelium, surface purplish pink to white (6A2/1A1), border grey (14B1); reverse purplish pink to grey (14A3/14 B 1), diffusible pigment absent. Colonies on MEA 2% reaching 18–20 mm diam., flattened, velvety, margin regular, greyish brown to dull red (8E3/8C3); reverse reddish brown to greyish red (8F7/7B3), diffusible pigment reddish brown (8D5). Colonies on OA reaching 30–32 mm diam., flattened to slightly floccose, margins regular, with sparse aerial mycelium, deep magenta to purplish grey, with greyish magenta patches (13D8/13D1/13D5), borders white; reverse deep magenta to olive grey with greyish magenta patches (14D8/1E2/13D5), diffusible absent. Cardinal temperatures of growth: Optimum 15–20 °C, maximum 28 °C, minimum 5 °C.

*Material examined:* Spain, Tarragona province, *Serra del Montsant*, from freshwater submerged plant debris, February, 2018, collected by Eduardo Jose de Carvalho Reis, holotype CBS H-24462, culture ex-type FMR 17,251 = CBS 146936.

*Notes: Murispora fissilispora*, genetically distinct from its neighboring *Murispora asexualis*, is the only species of the genus that produces both sexual and asexual morphs in vitro.

***Murispora asexualis*** V. Magaña-Dueñas, Cano & Stchigel, **sp. nov.** FMR 17,248. MycoBank MB 835711 (Figure 4).

*Etymology*. Because of the lack of a sexual morph, typical of the genus.

*Mycelium* composed of hyaline, smooth- and thin-walled, septate hyphae, 1.4–1.8 μm wide. *Conidiomata* pycnidial, solitary, brown to reddish brown, mainly immersed, glabrous, papillate, ostiolate, ovoid, 360–380 × 270–290 μm diam.; *peridial wall* of *textura angularis*, 4–6-layered, 20–40 μm thick, composed of brown to dark brown, flattened polygonal cells 3–4 μm diam.; *conidiophores* branched at the base, septate, hyaline to pale brown, straight or sinuous to slightly curved, 7.5–8.5 μm long; *conidiogenous cells* phialidic, hyaline, smooth- and thin-walled, ampulliform, slightly curved at the apex, 8–11 × 1–2 μm; *conidia* hyaline, non-septate, ovoid, 3–4 μm. *Sexual morph* unknown.

Culture characteristics (3 weeks at 15 °C). Colonies on PDA, reaching 30–32 mm diam., convex, velvety, margins irregular, with abundant aerial mycelium, surface reddish to white (12A2/1A1) margins grey (12C1); reverse violet brown to reddish brown (10E8/8D8), margins white, diffusible pigment absent. Colonies on MEA 2% reaching 24–28 mm diam., flattened, floccose, margins irregular, with abundant aerial mycelium, dark ruby to greyish ruby (12F3/12E6), margins reddish grey (12D2); reverse reddish brown to greyish red (8F7/7B3), diffusible pigment reddish brown (8D5). Colonies on OA reaching 38–42 mm diam., margins regular, mycelium mostly immersed, surface pink to yellowish white (12A4/4A2); reverse pink to yellowish white (12A4/4A2), diffusible pigment absent. Cardinal temperature for growth: Optimum 15–20 °C, maximum 30 °C, minimum 5 °C.

*Material examined*: Spain, Tarragona province, *Serra del Montsant*, from freshwater submerged plant debris, February, 2018, Eduardo Jose de Carvalho Reis, holotype CBS H-24463, culture ex-type FMR 17,248 = CBS 146937.

*Notes: Murispora asexualis* differs morphologically from the phylogenetically nearest species *M. fissilispora*, because it lacks a sexual morph. Furthermore, the conidiophores of *M. asexualis* are branched and slightly curved, while those of *M. fissilispora* are reduced to the conidiogenous cells.

***Murispora navicularispora*** V. Magaña-Dueñas, Stchigel & Cano, **sp. nov.** FMR 17,838. MycoBank MB 835712 (Figure 5).

*Etymology*. From Latin *navicularis*-, boat-shaped, and -*sporarum*, spore, because the shape of the ascospores.

*Mycelium* composed of hyaline, smooth- and thin-walled, septate hyphae, 1.4–1.8 μm wide. *Ascomata* perithecial, immersed to semi-immersed, solitary, brown to dark brown, ostiolate, papillate, *neck* conic-truncate, 100–90 × 60–70 μm, pyriform, 190–265 × 160–250 μm; *peridial wall* 2–4-layered, 20–50 μm thick, outer wall of *textura intricata*, composed of hyaline to brown hyphae 1.5–3 μm diam., inner wall composed by hyaline flattened cells; *hamathecium* comprising numerous hyaline, septate, filamentous, branched *paraphyses*, periphysate. *Asci* 8-spored, bitunicate, cylindrical to cylindrical-clavate, 115–120 × 15–20 μm, without apical structures. *Ascospores* 3–7-septate, cinnamon, broadly fusiform to navicular, 21–29 × 6–9 μm, narrowing towards the extremes, constricted at the septa, surrounded by a mucilaginous sheath. Natural substrate stained in purple. *Asexual form* unknown.

Culture characteristics (after 3 weeks at 15 °C). Colonies on PDA reaching 25–30 mm diam., umbonate, velvety, slightly cottony center, surface orange white to reddish white (5A2/6A2), pale orange (5A3) at the regular margins; reverse violet brown to yellowish white (10E4/4A2), diffusible pigment orange white (5A3). Colonies on MEA 2% reaching 26–28 × 17–20 mm diam, ellipsoidal, velvety, convex, white to reddish grey (8B2), with regular margins; reverse reddish brown (8F5), orange white (6A2) at the margins, diffusible pigment absent. Colonies on OA reaching 40–48 mm diam., flattened, with sparse aerial mycelium, surface and reverse deep violet (15E8), with yellowish white (4A2) regular margins; diffusible pigment absent. Cardinal temperatures of growth: Optimum 15–20 °C, maximum 30 °C, minimum 5 °C.

*Material examined*: Spain, Sevilla province, Cazalla de la Sierra, *Cascadas del Huéznar*, from freshwater submerged plant debris, May 2019, collected by José Francisco Cano Lira, holotype CBS H-24464, culture ex-type FMR 17838.

*Notes:* The fungus produces cinnamon, broadly fusiform to navicular ascospores, features never seen in the genus before.

*Lophiostomataceae* Sacc. Sylloge Fungorum, 2:672 (1883). MycoBanK 80966.

*Type genus: Lophiostoma* Ces. & De Not., Comm. Soc. crittog. Ital. 1(fasc. 4): 219 (1863). MycoBank MB 2933.

The genus *Pseudomassariosphaeria* was introduced by Phukhamsakda et al. in 2015 [17], to accommodate *Pseudomassariosphaeria bromicola,* found in a dead stem of *Bromus sterilis* L., transferring also *Massariosphaeria grandispora* to this genus (as *Pseudomassariospaheria grandispora*). However, in our phylogenetic study *P. bromicola* is clearly placed into the family *Amniculicolaceae* (transferred by us to the genus *Murispora* as *M. bromicola* earlier in this manuscript), whereas *P. grandispora* was located in the family *Lophiostomataceae,* phylogenetically close to *Lophiostoma macrostomum* and *L. arundinis.* The placement of *P. grandispora* into the *Lophiostomataceae* was previously suggested by Wang in 2007 [36], based on a molecular analysis using 28S rDNA, 18S rDNA and *rpb*2 gene. Consequently, we resurrected the name *Massariosphaeria grandispora* for this fungus.

*Massariosphaeria grandispora* (Sacc.) Leuchtm., Sydowia 37: 172 (1984). MycoBank MB 114956.

Description: Phukhamsakda et al. 2015.

*Basionym*: *Leptosphaeria grandispora* Sacc. Michelia 1(3): 341 (1878).

*Synonyms: Lophiotrema grandispora* (Sacc.) Shoemaker & C.E. Babc., Can. J. Bot. 67(5): 1580 (1989).

*Metasphaeria grandispora* (Sacc.) Sacc., Syll. fung. (Abellini) 2: 181 (1883).

*Neomassariosphaeria grandispora* (Sacc.) Y. Zhang ter, J. Fourn. & K.D. Hyde, in Zhang, Schoch, Fournier, Crous, Gruyter, Woudenberg, Hirayama, Tanaka, Pointing, Spatafora & Hyde, Stud. Mycol. 64: 96 (2009).

*Pseudomassariosphaeria grandispora* (Sacc.) Phukhams., Ariyaw. & K.D. Hyde, in Ariyawansa et al., Fungal Diversity: 10.1007/s13225-015-0346-5, [17] (2015).

## 4. Discussion

Of the three morpho-ecological groups of freshwater fungi (Ingoldian’s, aero-aquatic and facultative) only the latter two were addressed in this study. In our phylogenetic analysis, all of the *Amniculicolaceae* species clustered in a distinct sister clade to *Lindgomycetae*, which is similar to previous studies [16,17,18]. Most *Aminiculicolaceae* species are reported from freshwater habitats and are widely distributed across Austria, Italy, France, Germany, Denmark, China, Hungary and Spain [15,16,17,18,34,37]. However, with exception of *Murispora aquatica* and *M. triseptata* (basionym *Pseudomassariospaheria triseptata*)*,* all species of *Murispora* were isolated from terrestrial habitats such as dead terrestrial stems and dead and fallen twigs [14,15,16,17,18,35]. In this study, we have introduced three new species of *Murispora* collected from Spain in freshwater habitats. Thanks to the phenotypic characterization of several fungal isolates and to the subsequent phylogenetic analysis based on a concatenate database of the ITS-*LSU*-*tef1* sequences, we have erected three new species of *Murispora*: *M. asexualis,* the unique species of the genus because it lacks a sexual morph; *M. fissilispora,* the first species of this genus to produce a holomorph in vitro, and *M. navicularispora*, which produces cinnamon-colored, broadly fusiform to navicular ascospores, features never seen in the genus before. In addition, we have proposed the new combinations *M. bromicola* (formerly *P. bromicola*) and *M. triseptata* (formerly *P. triseptata*), demonstrating that this genus is monophyletic. Consequently, we have enlarged the current concept of *Murispora*, including species with hyaline, navicular and transversally septate ascospores, or lacking a sexual morph. Our results also indicate that some morphological features, such as the size and shape of the ascospores, have less phylogenetic significance than previously proposed by other authors. Despite *Spirosphaera cupreorufescens* displaying features considered as typical of that genus, it was phylogenetically distant in our phylogeny (in the class *Dothideomycetes*) from the type species of the genus (*Spirosphaera floriformis*, in the class *Leotiomycetes*), and because *S. cupreorufescens* formed a strongly supported clade together with two of our strains, we have proposed the erection of the new genus *Fouskomenomyces,* to include *Fouskomenomyces cupreorufescens* (the type species of the genus) and the new species *Fouskomenomyces mimiticus,* both aero-aquatic conidial fungi. Finally, we have also resurrected *Massariosphaeria grandispora*, because in our phylogeny it was placed into the *Lopiostomataceae* instead of the *Amniculicolaceae*. To date, there have been few reports of fungi isolated from freshwater habitats in Spain, therefore this work represents an important contribution to the knowledge of the Spanish mycobiota in aquatic environments. 

## Figures and Tables

**Figure 1 microorganisms-08-01355-f001:**
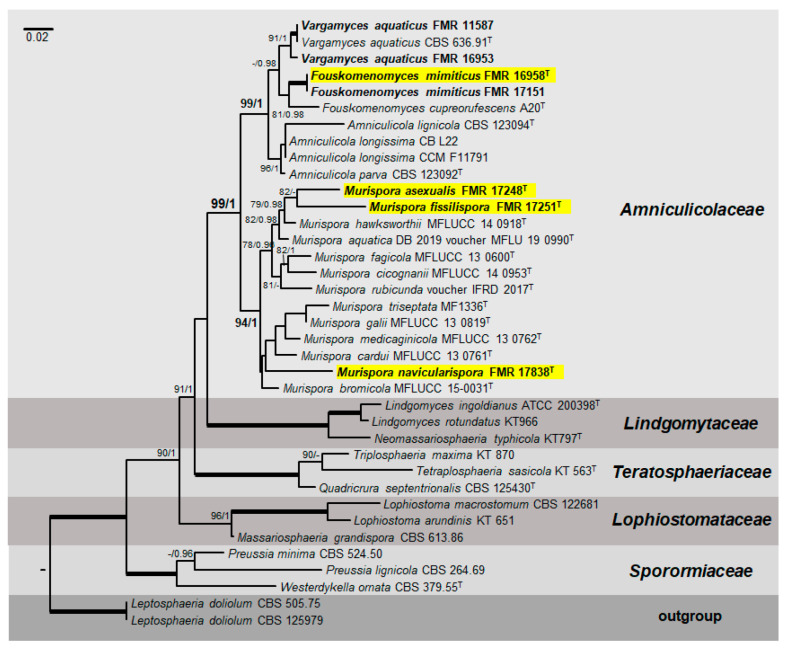
Phylogenetic tree inferred from a maximum likelihood analysis based on a concatenated alignment of D1-D2 domains of the 28S nrRNA gene (*LSU*), the internal transcribed spacer region (ITS) of the nrDNA, and a fragment of the translation elongation factor 1-alpha (*tef1*) gene sequences of 37 strains representing species in *Amniculicolaceae, Lindgomytaceae, Lophiostomataceae, Sporormiaceae* and *Teratosphaeriaceae.* The Bayesian posterior probabilities (PP) above 0.95 and the RAxML bootstrap support values (BS) above 70% are given at the nodes (PP/BS). Fully supported branches (1 PP/100 BS) are indicated in bold. Strains isolated during the developing of this work are in bold. Newly proposed taxa are highlighted in a yellow background. Type strains are indicated by a superscript “T”. The tree was rooted with *Leptosphaeria dolium* (CBS 125,979 and CBS 505.75). Alignment length 1,936 bp.

**Figure 2 microorganisms-08-01355-f002:**
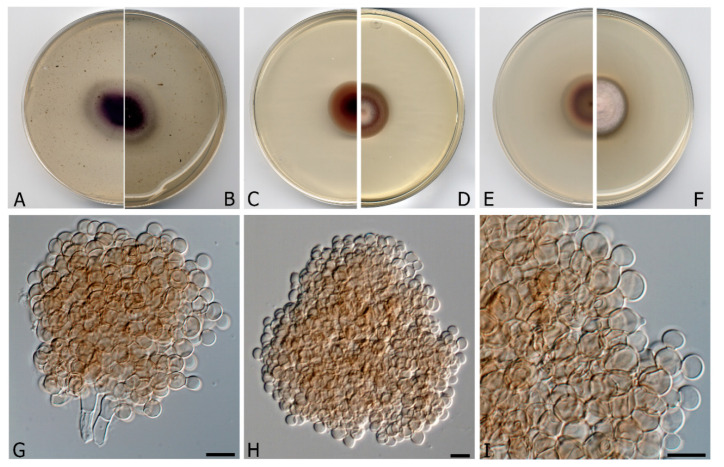
*Fouskomenomyces mimiticus* CBS H-24461. (**A**,**B**) Colonies on OA (oatmeal agar; reverse and front), (**C**,**D**) MEA (malt extract agar) 2% (reverse and front), (**E**,**F**) PDA (potato dextrose agar; reverse and front), at 20 °C after 3 weeks. (**G**) Conidial propagules with attached conidiogenous cells. (**H**) Free propagule. (**I**) Detail of a propagule showing the budding-like cells. Scale bars = 10 µm.

**Figure 3 microorganisms-08-01355-f003:**
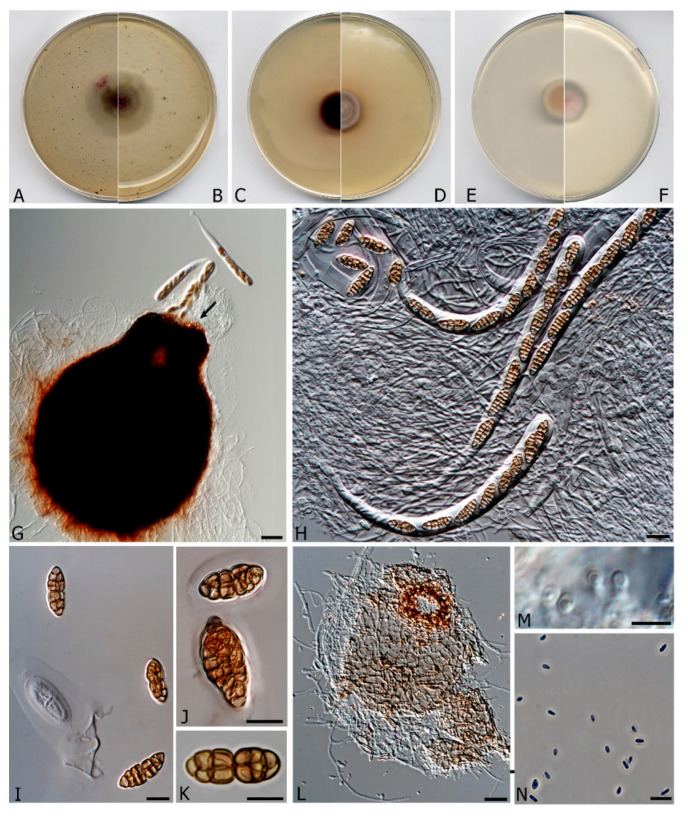
*Murispora fissilispora* CBS H-24462. (**A**,**B**) Colonies on OA (reverse and front), (**C**,**D**) MEA 2%(reverse and front), (**E**,**F**) PDA (reverse and front). All media at 15 °C after 3 weeks. (**G**) Ascomata expelling asci. (**H**) Asci. (**I**–**K**) Ascospores (Note the mucilaginous sheath in I and J). (**L**) Pycnidia. (**M**) Conidiogenous cells. (**N**) Conidia. Scale bars: G = 25 µm, H–N = 10 µm.

**Figure 4 microorganisms-08-01355-f004:**
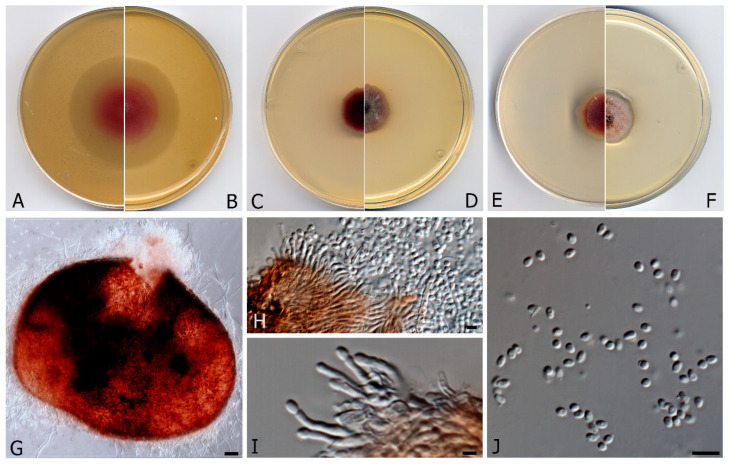
*Murispora asexualis* CBS H-24463. (**A**,**B**) Colonies on OA (reverse and front), (**C**,**D**) MEA 2% (reverse and front), (**E**,**F**) PDA (reverse and front). All media at 15 °C after 3 weeks. (**G**) Pycnidia (**H**,**I**) Conidiogenous cells (**J**) Conidia. Scale bars: (**G**) = 25 µm, (**H**,**J**) 10 µm, (**I**) = 2.5 µm.

**Figure 5 microorganisms-08-01355-f005:**
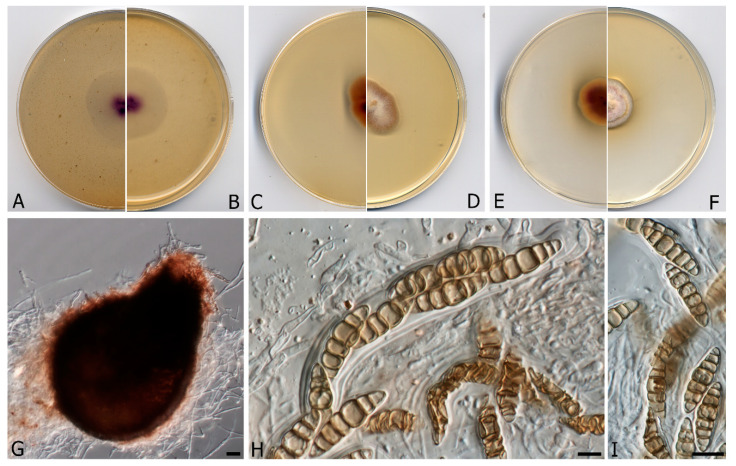
*Murispora navicularispora* CBS H-24464 (**A**,**B**) Colonies on OA (reverse and front), (**C**,**D**) MEA 2% (reverse and front), (**E**,**F**) PDA (reverse and front). All media at 15 °C after 3 weeks. (**G**) Ascomata. (**H**) Asci. (**I**) Ascospores. Scale bars: (**G**) = 25 µm, (**H**,**I**) = 10 µm.

**Table 1 microorganisms-08-01355-t001:** Fungal taxa and their nucleotide sequences of the molecular markers used to build the phylogenetic tree.

Taxon	Strain	GenBank Accession No.		
		*LSU*	ITS	*tef1*
***Amniculicola lignicola***	CBS 123094^T^	MH874798	-----	GU456278
*Amniculicola longissima*	CB L22	GU266240	AY204596	-----
*Amniculicola lonsissima*	CCM-F10304	JN673029	AY204594	-----
*Amniculicola parva*	CBS 123092^T^	FJ795497	MH863272	GU349065
***Fouskomenomyces cupreorufescens***	A20^T^	AY616236	AY616232	-----
***Fouskomenomyces mimiticus***	**FMR 16,958^T^**	**LR824585**	**LR824586**	**LR824584**
***Fouskomenomyces mimiticus***	**FMR 17,151 = CBS 146935**	**LR824588**	**LR824587**	**LR824589**
*Leptosphaeria dolium*	CBS 125979	JF740283	JF740208	-----
*Leptosphaeria dolium*	CBS 505.75	GQ387576	JF740205	GU349069
*Lindgomyces ingoldianus*	ATCC 200398^T^	AB521736	JF419898	-----
*Lindgomyces rotundatus*	KT966	AB521739	JF419901	-----
*Lophiostoma macrostomum*	CBS 122681	EU552141	EU552141	LC001753
*Lophiostoma arundinis*	KT 651	AB618999	JN942965	LC001738
***Massariosphaeria grandispora***	CBS 613.86	FJ795507	-----	GU349036
*Murispora aquatica*	MFLU 19-0990^T^	MN325075	MN325085	MN337969
***Murispora asexualis***	**FMR 17,248^T^ = CBS 146937**	**LR824596**	**LR824593**	**LR824590**
***Murispora bromicola***	MFLUCC 15-0031^T^	NG_059595	NR_164235	KT305999
*Murispora cardui*	MFLUCC 13-0761^T^	NG_059607	KT736082	KT709190
*Murispora cicognanii*	MFLUCC 14-0953^T^	NG_059609	NR_155381	MK109804
*Murispora fagicola*	MFLUCC 13-0600^T^	NG_060797	NR_155379	KT709188
***Murispora fissilispora***	**FMR 17,251^T^ = CBS 146936**	**LR824597**	**LR824594**	**LR824591**
*Murispora galii*	MFLUCC 13-0819^T^	KT709175	NR_154629	KT709189
*Murispora haswksworthii*	MFLUCC 14-091^T^	KT709180	NR_138414	KT709192
*Murispora medicaginicola*	MFLUCC 13-0762^T^	NG_059609	NR_155380	KT709191
***Murispora navicularispora***	**FMR 17,838^T^**	**LR824598**	**LR824595**	**LR824592**
*Murispora rubicunda*	IFRD 2017^T^	FJ795507	-----	GU456289
***Murispora triseptata***	MF1336^T^	MK411002	-----	-----
*Neomassariosphaeria typhicola*	KT797^T^	AB521747	JF419906	-----
*Preussia lignícola*	CBS 264.69	MH878448	-----	GU349027
*Preussia minima*	CBS 524.50	MH868263	MH856741	DQ677897
*Quadricrura septentrionalis*	CBS 125430	MH875152	NR_119402	-----
*Triplosphaeria máxima*	KT 870	AB524637	NR_119407	-----
*Tetraplosphaeria sasicola*	KT 563^T^	AB524631	AB524807	-----
*Vargamyces aquaticus*	CBS 636.9^T^	KY853539	NR_154471	-----
*Vargamyces aquaticus*	FMR 11587	KY853538	KY853475	-----
***Vargamyces aquaticus***	**FMR 16,953**	**LR812096**	**LR812095**	-----
*Westerdykella ornata*	CBS 379.55^T^	NG_057861	NR_103587	GU349021

^1^ A20: Hermann Volgmayr; ATCC: American Type Culture Collection, Virginia, USA; CB: Christiane Baschien; CBS: Culture collection of the Westerdijk Biodiversity Institute, Utrech, The Netherlands; CCM: Czech Collection on Microorganisms, Masaryk University, Faculty of Science, Brno, Czech Republic; FMR: Facultat de Medicina, Reus, Spain; IFRD: IFRDCC: Culture Collection, International Fungal Research & Development Centre, Chinese Academy of Forestry, Kunming, China; KT: Kazuaki Tanaka; MFLUCC: Mae Fah Luang University Culture Collection, Chiang Rai, Thailand. ^2^ Strains studied by us are indicated in bold. ^T^ Ex-type strain.

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
