# Peer review of "New Taxa of the Family Amniculicolaceae (Pleosporales, Dothideomycetes, Ascomycota) from Freshwater Habitats in Spain"

_microorganisms, 2020, doi:10.3390/microorganisms8091355_

Round 1
Reviewer 1 Report
The manuscript entitled “New taxa of the family Amniculicolaceae (Pleosporales, Dothideomycetes, Ascomycota) from freshwater habitats in Spain” is well written and the topic of interest. The materials and methods section is written with great detail. However, in my opinion, the paper needs few revisions according to the following comments before it can be published.
Page 7, in the caption of Figure 2, Colonies surface and reverse in the images A, B, C, D, E, F seem reversed. The sentence “(A, B) Colonies on OA (surface and reverse), (C, D) MEA 2% (surface and reverse), (E, F) PDA (surface and reverse), at 20°C after 3 weeks” could became “(A, B) Colonies on OA (reverse and front), (C, D) MEA 2% (reverse and front), (E, F) PDA (reverse and front), at 20°C after 3 weeks”.
Page 9, in the caption of Figure 3, the sentence “(A, B) Colonies on OA (front and reverse), (C, D) MEA 319 2% (front and reverse), (E, F) PDA (front and reverse)” could became “(A, B) Colonies on OA (reverse and front), (C, D) MEA 319 2% (reverse and front), (E, F) PDA (reverse and front)”.
Page 11, in the caption of Figure 4 change the sentence as suggested previously for the Figure 3.
Page 12, in the caption of Figure 5 change the sentence as suggested previously for the Figure 3.
Author Response
Dear Reviewer,
Thank you so much for the evaluation on our article. All changes suggested by you are accepted and introduced in the new version of the manuscript.
Best regards,
The authors
Reviewer 2 Report
Manuscript "New taxa of the family Amniculicolaceae (Pleosporales, Dothideomycetes, Ascomycota) from freshwater habitats in Spain" describes new isolated taxa of Ascomycota from freshwater, which is interesting because water fungi are not very well known. Authors erected new genus Fouskomenomyces. The scientific work is sound and solid within the realm of classical taxonomic studies. I think the study would gain more attention if the whole genomes would be sequenced and annotated.
Author Response
Dear Reviewer,
Thank you so much for the evaluation and comments on our article. We agree with you in the interest to sequence the whole genoma of such fungi. However, due to the limited of our economic funding, it could be one of objectives of a future reseach, which also could to includes the isolation and chemical structure elucidation of the diffusible pigments produced by these fungal taxa, which we also belive may have chemotaxonomic interest.
Best regards,
The authors